# Recent Developments in Chemical Derivatization of Microcrystalline Cellulose (MCC): Pre-Treatments, Functionalization, and Applications

**DOI:** 10.3390/molecules28052009

**Published:** 2023-02-21

**Authors:** Gabriele Lupidi, Genny Pastore, Enrico Marcantoni, Serena Gabrielli

**Affiliations:** Chemistry Division, School of Science and Technology, Università Degli Studi di Camerino, ChIP Building, Via Madonna delle Carceri, 62032 Camerino, Italy

**Keywords:** microcrystalline cellulose, pre-treatments, functionalization, applications

## Abstract

Microcrystalline Cellulose (MCC) is an isolated, colloidal crystalline portion of cellulose fibers, and it is a valuable alternative to non-renewable fossil-based materials. It is used for a large plethora of different fields, such as composites, food applications, pharmaceutical and medical developments, and cosmetic and material industries. The interest of MCC has also been driven by its economic value. In the last decade, particular attention has been driven to the functionalization of its hydroxyl groups to expand the field of applications of such biopolymer. Herein, we report and describe several pre-treatment methods that have been developed to increase the accessibility of MCC by breaking its dense structure allowing further functionalization. This review also collects the results that have appeared in the literature during the last two decades on the utilization of functionalized MCC as adsorbents (dyes, heavy metals, and carbon dioxide), flame retardants, reinforcing agents, energetic materials, such as azide- and azidodeoxy-modified, and nitrate-based cellulose and biomedical applications.

## 1. Introduction

Cellulose is the most widely available and abundant biopolymer in nature. It is regularly biosynthesized by plants, aquatic animals, fungi, and bacteria [1,2,3,4,5]. It is referred to as an inexhaustible feedstock regenerated by nature, with an annual fabrication estimated to be over 7.5 × 10^10^ tons [1]. It can be obtained both from top-down and bottom-up approaches. The first involves the extraction of cellulose from wood, cotton, plant, or agricultural residues, while the latter employs bacteria (e.g., *Acetobacter xylinum*) to biosynthesize cellulose from glucose [6]. Quantity, characteristics, and properties of the cellulose obtained depend on the extraction process and the nature of the material from which it is produced; commonly, cellulose extracted from biomasses may also contain hemicellulose, lignin in small quantities, and trace elements. Subsequent purification processes efficiently remove those impurities to obtain the final pure cellulose. Cellulose has a molecular formula of (C_6_H_10_O_5_)_n_ and is a linear carbohydrate polymer made of repeating glucose units connected by a 1–4 β glycosidic bonds. Every monomeric unit contains three hydroxyl groups, a primary one at C-6 in the methylol group (-CH_2_OH) and two secondary -OHs at C-2 and C-3. The presence of such functional groups is crucial for cellulose properties. Thanks to the capability of establishing hydrogen bonds and van der Waals interactions within the chain, the rigidity of the structure is preserved, avoiding rotation around the glycosidic bond. These are also responsible for the formation of ordered (crystalline) and disordered (amorphous) regions in the cellulose structure that significantly affect its chemical and physical behavior [7].

Cellulose can also be classified according to its crystalline structure as cellulose I, II, III, and IV (Figure 1). Cellulose I, or native cellulose, is the most abundant form in plant cell walls. We can find it in two distinct allomorphs: Iα consisting of a triclinic unit cell, and Iβ of a monoclinic unit cell. Cellulose produced by bacteria and algae is enriched in Iα, while cellulose in cotton, wood, and ramie fibers is predominantly Iβ [8,9,10]. The mercerized (alkali-treated) process can convert cellulose I into cellulose II. Cellulose III is prepared by treating native cellulose and cellulose II with anhydrous ethylamine or liquid ammonia and can be further converted into cellulose IV by glycerol treatments at high temperatures [11].

Furthermore, the three hydroxyl groups present in the glycosidic unit play a key role in its surface modification; thus, they can establish covalent bonds to several functional groups through different chemical modifications, providing additional and advantageous properties to the original matrix [13,14]. The derivatized cellulose can overcome some drawbacks of pure cellulose, increasing its solubility in water and common organic solvents. This advantage, combined with low cost, biocompatibility, and biodegradability, makes these derivatives particularly appealing for their applications in natural fiber-based polymer composites [15]; as thickeners, food stabilizers, gelling agents [16]; and for biomedical purposes [17]. Cellulose derivatives also find numerous applications in different areas; it is possible to tune their characteristics through chemical functionalization and, consequently, their properties and the activities they can carry out. For example, they find application in metal absorption with the possibility of removing contaminants from aqueous media, as already reported in several studies [18,19,20,21] or even as biosensors [22,23,24,25,26,27], and in the field of immobilization of enzymes [28,29]. For a more extensive reading on this topic, it is possible to check the review by Aziz et al. on the modifications of cellulose and its applications [30].

### Microcrystalline Cellulose

Microcrystalline Cellulose (MCC) is an isolated, colloidal crystalline portion of cellulose fibers, deriving from partial depolymerization of the cellulose matrix, conventionally by treating cellulose with an excess of mineral acid.

MCC can be co-processed with a water-soluble polymer to deliver a colloidal form or dried to a pure, fine-particle form (Figure 2).

The acid hydrolytic approach allows getting rid of the amorphous part of pristine cellulose fibers, whereas the ordered domains that present a higher resistance towards hydrolysis remain almost untouched [31,32]. Thus, this process reduces the degree of polymerization of cellulose chains, with a quite unmodified yield of the original matrix. Therefore, the MCC shows improved properties, such as high crystallinity index, large surface area, high density, excellent rigidity, and high thermal stability [33,34,35,36,37]. Furthermore, from a structural point of view, a large amount of accessible active hydroxyl moieties on the surface of MCC makes this matrix suitable for different chemical modifications, further enhancing its chemical and physical properties. These physico-chemical qualities, joined with great availability, easy preparation, odorlessness, and renewability, make microcrystalline cellulose a noteworthy candidate in many applications such as cosmetic, food, pharmaceutical, packaging, and polymer composite industries.

MCC in powder form is generally employed as a filler and binder in medical tablets and food, and as a reinforcement agent in polymer composites. On the other hand, in the colloidal form, it is used as a water retainer, suspension stabilizer, and emulsifier in different creams and pastes [7].

Nowadays, MCC is produced mainly from wood and cotton since they are the most abundant industrial sources of cellulose [38,39,40]. However, wood and cotton are only available in some areas. Competition between fields, such as paper industries, buildings, and furniture, makes these two sources only partially serviceable for cellulose and MCC production. Thus, interest in other sources, such as grass, aquatic and terrestrial plants, and agricultural wastes, has recently increased.

As mentioned, the most used process for the isolation of MCC is based on acid hydrolysis. It generally requires short reaction times and is relatively simple to be realized for a continuous production process. The cellulose matrix is composed of both crystalline domains and amorphous regions (also known as paracrystalline regions), which are preferentially hydrolyzed after contact with mineral acids. The temperature of the hydrolytic process, time, type of acid used, and concentration are crucial for the resulting physicochemical, thermal, and mechanical properties of the outcoming MCC. Research also reported how the fiber-to-acid ratio is a key feature to be considered since it plays a significant part in influencing the particle size, morphology, crystallinity, and thermal stability of MCC [41,42,43].

Trache et al. extensively reported other techniques to isolate microcrystalline cellulose in addition to classic acid hydrolysis [7]. In fact, Trusovs claimed a method to treat cellulose and produce MCC by oxidative degradation. This treatment is one of the most used methods for the delignification of cellulosic materials, where an alkaline solution is directly mixed with a source of cellulose. After the matrix is swollen, hydrogen peroxide is added to depolymerize the cellulose, and MCC is obtained after filtration and neutralization. Of note, this process does not require a high temperature and pressure and can be performed efficiently using commercially available chemicals with a relatively simple and economical method [44]. However, this procedure may lead to unwanted degradation of cellulose, thus making it impossible to isolate intact MCC.

In the late 90s, DeLong was able to obtain MCC treating a cellulose source material on a two-step process: (i) a steam explosion treatment to produce a low degree of polymerization cellulose, and (ii) treatment of the resulting material with a strong mineral acid, through which it was possible to recover microcrystalline cellulose [45].

Afterwards, Ha and Landi claimed a one-step steam explosion method for producing MCC directly from a source of cellulose. Under controlled reaction conditions, they made this matrix undergo a steam explosion process that allowed it to remove hemicellulose and lignin and recover microcrystalline cellulose in particle size, free from fibrous cellulose [46].

Reactive extrusion has recently been presented as an alternative and effective method to produce MCC from lignocellulosic residues [47]. It takes advantage of high-temperature hydrolysis methods. Essentially, the raw material is charged in an extruder, and a two-step separation is performed with NaOH first, followed by acidic treatment (H_2_SO_4_). Therefore, the first basic treatment allows the separation of the original lignocellulosic material into lignin, hemicellulose, and cellulose. Hemicellulose and lignin are removed from the obtained pulp. The remaining material is hydrolyzed by treatment with a mineral acid to form MCC quickly and efficiently as short, rod-shaped fibers with a high cellulose content and almost 70% crystallinity.

After MCC’s production and isolation, chemical derivatization can convert it into functional materials. However, its crystalline structure, composed of cellulose molecular chains arranged in an orderly manner through hydrogen bonding, makes microcrystalline cellulose less accessible to functionalization [48]. In order to enhance its reactivity, in recent years, attention has been driven to its activation.

## 2. MCC Pre-Treatment

Several pre-treatment methods have been developed to increase the accessibility of MCC by breaking its dense structure. Microcrystalline cellulose pre-treatments can be divided into physical, biological, and chemical processes.

### 2.1. Physical Pre-Treatment

This method does not require the assistance of any solvent, and it can destroy the crystalline structure of MCC by mechanical action. Lan et al. pre-treated microcrystalline cellulose with ball-milling, employing different treatment times (0.5–24 h), in order to modify its physiochemical properties and improve its activity in photoreforming over Pt/TiO_2_ for H_2_ production. In particular, the BM treatment reduced the degree of polymerization (DP), the particle size, and the crystallinity index (CrI) of MCC, as well as producing amorphous cellulose [49].

Yu et al. reported comparable results in their work, exploiting an innovative ball-milling process allowing MCC hydrolysis in hot-compressed water. An increase in MCC’s reactivity and a significant reduction in cellulose crystallinity and particle size was observed. They also found that the crystalline phase is wholly converted into amorphous regions with long-term pre-treatment. Using a cryogenic milling action (2 min) led to considerable size reduction but little change in reactivity and cellulose crystallinity. Furthermore, ball-milling plays a key role in the distribution and selectivity of glucose oligomers in the primary liquid products of MCC hydrolysis [50].

Ball-milling pre-treatment can also be used in combination with catalysts, as reported by Ribeiro et al. This work focused on a Ruthenium supported on activated carbon, as a catalyst, for the one-pot hydrolytic hydrogenation of cellulose. When Ru/Carbon was ball-milled together with MCC, the conversion and the selectivity to sorbitol increased. Moreover, in the one-pot conversion of microcrystalline cellulose, a decrease in the DP was observed but was not as significant as the decrease in the CrI [51]. Strong acids or solid catalysts must be employed to obtain a high depolymerization degree of MCC. Meine et al. showed that the ball-milled cellulosic materials impregnated with catalytic amounts of strong acids (HCl, H_2_SO_4_, etc.) are fully converted into water-soluble oligosaccharides within 2 h [52]. The same depolymerization products of cellulose can be obtained with the mechanocatalysis process, using ball-milling in combination with solid catalysts (kaolinite), as reported by Hick et al. [53].

Mechanocatalytic depolymerization of cellulosic materials is a one-pot process that combines mechanical action with acid catalysis to convert cellulose into water-soluble products. Upscaling the process from 1 g to 1 kg significantly reduces energy consumption (from ca. 200 MWh·t^−1^ to 9.6 MWh·t^−1^*)* [54]. These results, combined with the low environmental impact of this pre-treatment, make ball-milling a sustainable method to increase the accessibility of MCC [55,56,57].

Using acidic conditions to depolymerize MCC requires laborious working-up procedures, limiting the process’s scale-up. In fact, the acid hydrolysis step produces a large amount of waste liquid, mainly the added acid, along with reducing sugars and other cellulose degradation components [58]. To overcome this problem, Benoit et al. used the non-thermal atmospheric plasma in combination with ball milling as the physical pre-treatment to convert MCC into low molecular weight cellulose oligomers (DP = 36). Ball-milling treatment reduced cellulose crystallinity, while the plasma process induced the depolymerization of MCC without using any catalyst or solvent. Furthermore, they found that the non-thermal atmospheric plasma is highly selective since degradation and oxidation, which are competitive processes occurring on plasma-treated materials, occur to a very small extent within 1 h of treatment. The oligomers obtained are more prone to hydrolysis than native cellulose. However, as reported, the reaction mechanism is complicated to clarify fully [59].

Huang et al. used dielectric barrier discharge plasma as a pre-treatment method to depolymerize MCC. The decrease in the DP was the same when argon or air was used as the carrier gas. After 30 min of DBD, the liquefaction yield and the total reducing sugar concentration increased compared to those of raw microcrystalline cellulose [60].

Among physical pre-treatments, ultrasound technology is considered a green approach, where chemical (oxidizing radicals) and physical mechanisms (shear forces, cavitation) are involved. In particular, the energy provided by cavitation is within the hydrogen bond energy levels and may induce both reorganizations in the amorphous phase of cellulose and defibrillation. Furthermore, ultrasound was used for promoting many organic reactions [61], obtaining novel cellulose materials [62], extracting lignin and hemicellulose from lignocellulosic materials [63], and dissolving cellulose [64].

Vizireanu et al. reported using ultrasound coupled with submerged liquid plasma to modify MCC [65] and Zhang et al. show that ultrasound pre-treatment can enhance the reactivity of microcrystalline cellulose under heterogeneous acidic catalysis. A significant decrease in particle size and crystallinity was observed with better interaction between the solid catalyst and MCC. Moreover, they found that ultrasound as a pre-treatment gives comparable results with respect to ball-milling and ionic liquids. Additionally, water is used as a unique solvent, MCC is not contaminated with pollutants, short reaction times are required (3 h), and the heating ensured by the energy dissipated by this pre-treatment contributes to decreasing the energy consumption of the whole process [66]. 

The last method, reported by Zhang et al., combines two different processes and demonstrates that ultrasound technology can also be combined with the Fenton reagent as pre-treatment of MCC for subsequent depolymerization by enzymatic hydrolysis. X-ray diffraction and DP analyses showed that the Fenton reagent was more efficient in decreasing MCC’s crystallinity, while ultrasound decreased its DP. Thus, combined pre-treatment processes for cellulose depolymerization can be considered an innovative and promising approach to overcome drawbacks exhibited by various individual pre-treatment processes [67].

### 2.2. Biological Pre-Treatment

This method is environmentally friendly and involves enzymes to hydrolyze MCC, but it is less exploited due to the longer reaction times required to achieve significant conversion [68]. Furthermore, using chemical pre-treatments, such as mercerization in concentrated sodium hydroxide, is necessary in order to enhance the enzyme activity on microcrystalline cellulose, as reported by Chen et al. They studied the molecular weight and crystallinity of pre-treated MCC digested with three pure *Thermobifida fusca* Cellulases [69].

### 2.3. Chemical Pre-Treatment

Most of the chemical methods reported in the literature involve substances that can drastically reduce reaction times and are more advantageous than energy-intensive physical technologies and relatively low-efficient biological pre-treatment.

One of the most known chemical pre-treatments is mercerization. This is the treatment of cellulose under basic conditions, such as NaOH, NaOH/Urea, or LiOH/Urea aqueous solutions, where the native cellulose is converted into alkali cellulose I and II. However, this process is less exploited as a pre-treatment because it produces much basic solution waste [70,71].

Among chemical pre-treatments, the dissolution of MCC in ionic liquids (ILs) is one of the most promising. The intermolecular hydrogen bonds make cellulose poorly soluble in conventional solvents, except for ILs, which can break these bonds and increase the accessibility of cellulose to further reactions [72].

To date, there are two points of view concerning the interface relationship between ILs and cellulose: (i) the driving force for cellulose dissolution concerns the -OH hydrogen bonds formed with the anion and the cation of ILs; (ii) the anion and the carbohydrate interaction rule the dissociation process, and there is no specific relationship between cellulosic materials and the cation. Some experimental results have supported each of these views.

Zhao et al. applied molecular dynamics simulation to study cellulose dissolution in the ILs with the same anion (Cl^−^), but different cations. The ILs used include 1-alkyl-3-methyl imidazolium ILs [C_n_mim]Cl (n = 2, 3, 4, 6, 8, 10), 1-butyl-3-methyl pyridinium chloride [C4mpy]Cl and 1-allyl-3-methyl imidazolium chloride [Amim]Cl. The structures for these ILs are shown in Figure 3 [73].

The results indicate that the cations’ heterocyclic structure and alkyl chain length have an important influence on the interactions between the anion of the IL and cellulose.

On the other hand, ILs constituted by anions, which are strong hydrogen bond acceptors, are more effective, especially when combined with microwave heating. For example, ionic liquids containing non-coordinating anions, such as [BF_4_]^−^ and [PF_6_]^−^, are non-solvents, while chloride-containing ionic liquids dissolve MCC [74]. For this purpose, different chloride-containing ionic liquids were evaluated as pre-treatment for microcrystalline cellulose. Both authors, however, report that the presence of long alkyl chains in the cations may increase the steric hindrance of the anion to bind with cellulose, thus decreasing the dissolution ability of the ILs for cellulose.

ILs are also employed to treat biomasses, as reported by Liu et al. They found that 1-butyl-3-methylimidazolium chloride ([BMIM]Cl) improves the enzymatic hydrolysis of steam-exploded wheat straw and wheat straw. In particular, a reduction in the crystallinity and polymerization degrees of cellulose and hemicellulose was observed [75].

Berga et al. found that the maximum dissolution of MCC was obtained in a 60:40 molar ratio of 1,1,3,3-tetramethylguanidine and propionic acid [TMGH][OPr], respectively. Altering this ratio towards an acid-rich composition leads to the regeneration of microcrystalline cellulose, exploiting the antisolvent nature of propionic acid. However, the use of ILs on an industrial scale has some drawbacks, such as the high cost, the potential hazards for human beings and the environment, the difficulty in the purification of ionic liquids, and the complete removal from microcrystalline cellulose [76]. Furthermore, the presence of residual ILs, even in trace amounts, results in a significant deactivation of catalysts or enzymes used in the subsequent hydrolysis of cellulose [77,78].

Other solvents for microcrystalline cellulose dissolution were evaluated, such as LiCl/DMAc (lithium chloride with *N*,*N*-dimethylacetamide as solvent), as reported by Wang et al. In particular, the chloride anions interact with the hydroxyl groups of MCC, leading to cellulose chains’ solvent infiltration and swelling. The accessibility of hydroxyl groups (-OH) in native cellulose is very low, making its dissolution hard in a reasonable time frame. For this reason, MCC must be pre-activated with other solvents such as water, methanol, or DMAc.

Dimethylacetamide and water cause the exfoliation of cellulose after a long time of soaking, and MCC could be further dissolved in LiCl/DMAc. DMAc promotes the nano-pore formation of microcrystalline cellulose, which favors the diffusion of chloride anions into cellulose chains and the dissolution of MCC. On the other hand, water pre-activation prevents the diffusion of chloride anions due to a strong nano-pore closing effect, especially for pores with a diameter of less than 30 nm [79].

Jin et al. found that untreated cellulose can be dissolved quickly and directly in a NaOH/thiourea/urea aqueous solution, where cellulose I is converted into amorphous cellulose and then to cellulose II. This system possesses a higher MCC solubility capacity than NaOH/urea and NaOH/thiourea aqueous solutions. They also investigated the mechanism for dissolution, showing that the interactions between NaOH-thiourea and NaOH-urea play a key role. Moreover, ^13^C-NMR spectra proved that NaOH, thiourea, and urea were bound to cellulose molecules, bringing them into the aqueous solution and preventing their association [80].

Effective cellulose solvents are also represented by molten salt hydrate due to their high chemical and thermal stability, low viscosity, and recyclability. Huang et al. reported a non-dissolving pre-treatment of cellulose with lithium bromide trihydrate (LBTH) under ambient conditions that led to the decrystallization and deconstruction of cellulose. The decrystallization process began once cellulose interacted with LBTH, and after five minutes, the crystalline structure of MCC was completely transformed to amorphous. After 30 min of LBTH pre-treatment, MCC was completely deconstructed, and the accessibility of cellulose was greatly improved. Furthermore, this pre-treatment under ambient conditions without dissolution of cellulose makes a recovery and reuse of lithium bromide trihydrate much easier to handle. This solvent could be recovered by phase separation and distillation without losing its ability to deconstruct and decrystallize MCC. Therefore, this effective pre-treatment has very low energy input, which can very efficiently improve the accessibility of cellulose for further reactions [81].

Inorganic salt hydrates are a good alternative solvent for microcrystalline cellulose pre-treatment, as reported by Lara-Serrano et al. They found that the dissolution of microcrystalline cellulose in different inorganic salt hydrates (ZnCl_2_·4H_2_O, ZnBr_2_·4H_2_O, LiCl·8H_2_O and LiBr·4H_2_O) was very fast (15 min) with a temperature of 70 °C. The XRD and SEM analysis confirmed that the change in microcrystalline cellulose morphology increased the rate of hydrolysis with respect to that of untreated cellulose. Furthermore, bromide salt hydrates (ZnBr_2_·4H_2_O and LiBr·4H_2_O) are more efficient and dissolve cellulose faster than their chloride counterparts. Generally, salt hydrates can be reused after the evaporation of the excess water used in the precipitation and washing of the treated samples [82].

## 3. MCC Functionalization and Applications

In the last few years, the use of microcrystalline cellulose in functional components, such as MCC-fibres reinforced materials or exploiting its chemical functionality in order to modify the structure, has become an attractive field of research activity. The chemical derivatization of MCC with functional components leads to the formation of advanced materials with new or improved properties. Therefore, their functionalities will be improved, and potential applications in specific fields can be expanded. Functionalized MCC can be used as adsorbents (dyes, heavy metals, and carbon dioxide), flame retardants, reinforcing agents, energetic materials, and biomedical applications. Certainly, MMC, similar to all other cellulose derivatives, finds other useful applications in the field of cellulose-based lubricants, thanks above all to its outstanding dry-binding properties. However, due to the instability of the dispersion of chemical derivatization of MCC, they cannot be utilized directly as lubricants. Such dispersions can be made stable by adding a third component that can create a structural network to prevent aggregation and sedimentation of the MCC particles [83]. For this reason, these applications as lubricants are not taken into consideration in this review.

### 3.1. Adsorbents

Microcrystalline cellulose usually exhibits low adsorption ability toward dyestuffs. Therefore, its functionalization is necessary to obtain system with improved adsorption ability, as reported by Wei et al. They studied the functionalization of microcrystalline cellulose aerogel and its absorption efficiency towards dyes such as methylene blue. For this purpose, MCC was coated with polydopamine (PDA), which was realized via the self-polymerization of dopamine in the MCC/LiBr solution, followed by the freeze-drying step. The MCC/PDA system shows high adsorption efficiency and selectivity towards methylene blue from different solutions (mixed solution with common salts, cationic and anionic dyestuffs). Moreover, the high structural stability of the microcrystalline cellulose network makes the aerogel stable in aqueous conditions, and it has great potential application in wastewater treatment to reduce the emission of such contaminants, hence decrease water pollution [84].

Other systems based on MCC can be used as adsorbents of dyes. Hu et al. reported that microcrystalline cellulose functionalized with quaternary ammonium groups, using *N*,*N*-dimethyldodecylamine as nucleophile onto glycidyl moieties, is able to remove the Congo Red dye (CR) from aqueous solution. Ultrasound was used as a pre-treatment to disrupt the original structure of microcrystalline cellulose and make hydroxyl groups more accessible, as this is a two-step functionalization (Figure 4).

The adsorption of Congo Red dye depends on the solution’s pH, dye concentration, NaCl concentration, and temperature. The maximum adsorption capacity of MCC functionalized with *N*,*N*-dimethyldodecylamine reached 304.34 mg·g^−1^ (with a dye concentration of 80 mg·L^−1^, a volume of 100 mL, a temperature of 40 °C, and an adsorbent dosage of 10 mg) [85].

Tannic acid (TA) functionalized MCC was used for the selective recovery of gallium (Ga) and indium (In), as reported by Du et al. The functionalization was performed by radiation-induced grafting polymerization. The selective separation and recovery of Ga (III) and In (III) from potential leaching solutions were investigated using batch and fix-bed column experiments. Firstly, a radiation grafting technique prepared glycidyl methacrylate-grafted cellulose (MCC-g-GMA). The MCC-g-GMA was added to a TA aqueous solution to obtain the MCC-g-GMA-TA system (Figure 5). The functionalized MCC can recover Ga (III) from the simulated semiconductor waste-leaching solution containing As (III), Cu (II), Zn (II), and Ni (II) with a maximum adsorption capacity of 26.55 mg·g^−1^. On the other hand, In (III) can be separated from simulated Zn refinery residue leaching solution containing Cu (II), Zn (II), and Ni (II) with a maximum adsorption capacity of 35.63 mg·g^−1^. The FTIR, XPS analyses, and the ionic strength effect revealed that functionalized MCC can selectively recover gallium and indium by forming chelates between the hydroxyl groups and Ga^+3^ or In^+2^ [86].

Microcrystalline cellulose can also be used to improve heavy metal adsorption from water and CO_2_ uptake. For this purpose, Rafieian et al. performed two successive grafting reactions for the chemical modification of MCC for the removal of heavy metals from industrial wastewater. The first reaction is the silylation by (3-chloropropyl)triethoxysilane (CPTES), and the second is the amine-functionalization with 1,1-dimethylbiguanide hydrochloride (Figure 6). The grafting of CPTES onto MCC surfaces involves the hydrolysis of the alkoxy groups of the silane and the formation of hydrogen bonds between the silanol obtained and the hydroxyl groups of cellulose with consequent adsorption and grafting onto -OH on the surface of MCC. Finally, the chemical condensation and the formation of a polysiloxane network on the surface of microcrystalline cellulose were performed [87].

On the other hand, Gunathilake et al. synthesized different cyanopropyl-incorporated microcrystalline cellulose-organosilica (MCC-CP) mesostructured materials in a two-step process for carbon dioxide sorption. They first performed a solvent evaporation-induced self-assembly of microcrystalline cellulose, tetraethyl orthosilicate, and (3-cyanopropyl)triethoxysilane in the presence of a triblock copolymer under acidic conditions. Cyanopropyl groups were further converted into amidoxime functionalities upon treatment with hydroxylamine hydrochloride (Figure 7). Different amidoxime-functionalized microcrystalline cellulose-mesoporous silica composites (MCC-AO) were obtained, and their CO_2_ sorption at ambient (25 °C) and elevated (120 °C) temperatures was evaluated. The composites showed low CO_2_ uptake at 25 °C, and sorption capacities between 2.15–3.85 mmol·g^−1^ are reached with high temperatures (120 °C). The values obtained confirm that MCC’s amine, oxime, and hydroxyl groups are active sites for CO_2_ sorption. Furthermore, the selectivity of these composites towards carbon dioxide in the presence of nitrogen, in addition to their low cost, high thermal and mechanical stability, non-toxicity, biodegradability, and biocompatibility, makes them potential candidates for CO_2_ sorption at elevated temperatures [88].

### 3.2. Flame Retardants

Functionalized microcrystalline cellulose can be also used as a flame retardant. In particular, Yuan et al. reported the chemical modification of MCC with phytic acid (PA) by phosphorylation reaction for enhanced flame-retardant ability. The modified microcrystalline cellulose was synthesized at 90 °C with a 1:3 weight ratio of MCC to PA (Figure 8). The resulting system showed low heat release performance and good char-forming ability during thermal degradation. This work reveals that the PA phosphorylated MCC can act as a promising flame-retardant material [89].

Finally, Lou et al. studied and synthesized a fully bio-based flame retardant with high efficiencies and mechanical reinforcement functions for epoxy resin. Microcrystalline cellulose was surface functionalized with chitosan (CS) and sodium phytate (PA-Na) via layer-by-layer assembly in water (Figure 9). The functionalized MCC catalyzes the degradation of epoxy resin to produce char residues with the formation of stable phosphate and nitride species in the condensed phase. The char can act as a physical barrier to inhibit smoke production and heat transfer. This result shows that the total heat release, total smoke production, peak heat release rate, fire growth rate, peak smoke production release, and fire retardancy index of the epoxy composites are greatly reduced, confirming their good fire-retardant ability. Moreover, adding 15 wt.% of modified microcrystalline cellulose can simultaneously enhance the composites’ strength, modulus, and toughness due to the satisfactory interfacial compatibility, favorable dispersion, and mechanical reinforcement effect of MCC [90].

### 3.3. Reinforcing Agent

Thanks to its renewability, biodegradability, low cost, and low density, MCC is considered a new class of cellulosic reinforcing agent that improves the sustainability of the final composite compared to silica, carbon black, and glass fibers.

Ashori et al. reported the use of microcrystalline cellulose as a reinforcing agent in polypropylene (PP)/microcrystalline cellulose (MCC)/wood flour composites containing polypropylene-graft-maleic anhydride (PP-g-MA) as a compatibilizer.

The weight ratio of the cellulosic materials to polymer matrix was 40:60 (*w*:*w*), and the final mechanical properties (flexural, tensile, and impact strengths) were significantly enhanced with the addition of microcrystalline cellulose, as compared with pure PP and composites without MCC. Scanning electron microscopy confirms that using maleic anhydride as a compatibilizer increases the fiber–matrix interaction. Moreover, the addition of PP-g-MA increases the degradation temperatures of composites with an improvement in thermal stability, which reaches the highest value when 5 wt.% of polypropylene-graft-maleic anhydride was used [91].

Ratnakumar et al. reported that microcrystalline cellulose acts as a reinforcement for polypropylene. They examined the properties of MCC-reinforced polypropylene composites varying the MCC loading from 0–5%. The final composite shows improved properties as the MCC loading increases. Tensile, hardness, and impact properties were improved by 5%, 8%, and 51% for 5 wt.% MCC-reinforced PP composite [92].

Microcrystalline cellulose was also used as a reinforcing agent in polydimethylsiloxane (PDMS)/MCC composites in order to improve the mechanical properties, as reported by Jankauskaite et al. Silver nanoparticles were synthesized in situ by chemical reduction method in MCC aqueous suspension to provide antimicrobial activity. Vinyl-terminated PDMS was mixed with MCC and silver nanoparticles, which combines stiffness and antimicrobial properties. However, voids around MCC demonstrate poor adhesion interaction at the PDMS matrix and MCC particle interface, as shown from SEM analysis in Figure 10.

Mechanical properties are improved compared to unfilled PDMS, and the final composite inhibited the growth and multiplication of the Gram-positive *Staphylococcus aureus* and Gram-negative *Escherichia coli* bacteria after 1 h of incubation. However, the tendency of silver nanoparticles to aggregate and its low concentration led to bacteriostatic activity alone [93].

Deng et al. also reported the use of MCC as a reinforcing agent in silicones. Microcrystalline cellulose fibers from cotton linter were modified with propargyl bromide in an aqueous medium. The functionalized MCC was introduced as filler in silicone fluids, followed by crosslinking at high temperatures between hydride-functional and vinyl-functional PDMS to produce silicone elastomer composites. Furthermore, in the hydrosilylation process, it was expected to be also involved in the alkyne functionalized MCC (Figure 11). The presence of alkyne functions favors the interfacial adhesion between silicone and MCC and the densification of the network. The resulting composite showed an enhancement of mechanical and thermal properties when MCC-Alkyne was incorporated as filler, compared to unmodified MCC [94].

### 3.4. Energetic Materials

Among cellulose-based energetic materials, nitrocellulose is the most used in double-base rocket propellants and gunpowder. However, new alternatives have been investigated in the last years due to the high friability, low combustion temperature, chemical instability, and high sensitivity of nitrocellulose.

Tarchoun et al. reported the azide-modification of microcrystalline and pristine cellulose as promising nitrogen-rich energetic biopolymers. Azidodeoxy-microcrystalline cellulose nitrate (AMCCN) and azidodeoxypristine cellulose nitrate (APCN) were synthesized, and their properties were compared to cellulose nitrate (PCN) and microcrystalline cellulose nitrate (MCCN). PC and MCC were dissolved in DMAc/LiCl, and p-toluenesulfonyl chloride (hydroxyl activating agent), triethylamine (base catalyst), and subsequently sodium azide was added to obtain the corresponding azidodeoxy-pristine cellulose (APC) and azidodeoxy-microcrystalline cellulose (AMCC). Finally, the nitration of free-accessible hydroxyl groups of chemically functionalized and non-functionalized samples was performed with a mixture of fuming nitric acid and acetic anhydride to afford the corresponding PCN, MCCN, APCN, and AMCCN (Figure 12).

The newly synthesized AMCCN and APCN present high nitrogen contents, increased densities, and good thermal stabilities compared to the conventional PCN and microcrystalline cellulose nitrate. All nitrated polymers are insensitive against impact and friction, except for the nitrated unmodified ones (PCN and MCCN), which showed very high impact sensitivity. Furthermore, the experimentally measured heats of combustion and the predicted energetic performances of the investigated nitrated biopolymers indicated that the nitrated azide-modified biopolymers (APCN and AMCCN) displayed high heats of combustion, increased detonation velocities, comparable detonation pressures, low temperatures of explosion, and moderate specific impulses with respect to those of nitrated unmodified ones (PCN and MCCN) [95].

Tarchoun et al. also studied 6-deoxy-6-(ω-aminoethyl)nitramino microcrystalline cellulose nitrate (also referred to as microcrystalline cellulose amine nitrate, MCCAN) and 6-deoxy-6-(ω-aminoethyl)nitramino cellulose nitrate (also referred to as cellulose amine nitrate, CAN) as a new class of insensitive energetic nitrogen-rich polymers through chemical modification of cellulose and microcrystalline cellulose (MCC). Cellulose and MCC were dissolved in DMAc/LiCl, and p-toluenesulfonyl chloride with triethylamine was added to obtain cellulose tosylate (CT) and microcrystalline cellulose tosylate (MCCT). Then, the displacement of tosyl groups with ethylenediamine moieties was carried out in DMSO. Finally, 6-deoxy-6-(ω-aminoethyl)nitramino cellulose nitrate (CAN) and 6-deoxy-6-(ω-aminoethyl)nitramino microcrystalline cellulose nitrate (MCCAN) were synthesized with a mixture of fuming nitric acid and acetic anhydride (Figure 13).

The new synthesized energetic polymers (MCCAN, CAN) show good thermal stability, elevated densities, and high nitrogen content compared to cellulose nitrate and microcrystalline cellulose nitrate.

All nitrated polymers are insensitive to friction, whereas the aminated and nitrated polymers show extraordinarily lower impact sensitivities than those of unmodified polymers nitrate. The newly prepared CAN and MCCAN present high energy of the explosion, low detonation temperatures, comparable detonation pressures, high detonation velocities, and moderate specific impulses compared to the common cellulose nitrate and MCCN, respectively [96].

Finally, Tarchoun et al. reported the tetrazole-acetate modification of pristine cellulose (PC) and microcrystalline cellulose, using 1H-tetrazole-5-acetic acid as a functionalized agent. MCC and PC were dissolved in DMAc/LiCl, and p-toluenesulfonyl chloride with 1H-tetrazole-5-acetic was added to obtain 2-(1H-tetrazol-5-yl)acetate pristine cellulose (TAPC) and 2-(1H-tetrazol-5-yl)acetate microcrystalline cellulose (TAMCC). Finally, the nitration of OH groups of chemically modified and non-functionalized samples was carried out with a mixture of fuming nitric acid and acetic to afford the corresponding pristine cellulose nitrate (PCN), microcrystalline cellulose nitrate (MCCN), 2-(1H-tetrazol-5-yl)acetate pristine cellulose nitrate (TAPCN), and 2-(1H-tetrazol-5-yl)acetate microcrystalline cellulose nitrate (TAMCCN) (Figure 14).

Additionally, in this case, the newly synthesized 2-(1H-tetrazol-5-yl) acetate pristine cellulose and microcrystalline cellulose showed high nitrogen contents, increased densities, and good thermal stability compared to the conventional pristine and MCC-based cellulose nitrates. The friction sensitivities of functionalized nitrated polymers were lower than those of non-functionalized nitrated polymers. Furthermore, the new tetrazole-acetate modified MCC and PC show low heats of the explosion, decreased detonation temperatures, similar detonation pressures, high detonation velocities, and moderate specific impulses compared to pristine cellulose nitrate and microcrystalline cellulose nitrate [97].

## 4. Biomedical Applications

The biomedical applications of MCC are increasing. In fact, MCC showed a cytotoxic potential against normal fibroblasts, melanoma, and breast cancer [98], and no cytotoxicity in a hemolytic assay [99].

Kundu et al. synthesized three MCC-based hydrogels for the loading and in vitro release of Cephalexin, a semisynthetic antibiotic. MCC gel, MCC-CMC gel, and MCC-xylan gel were synthesized using ethylene glycol diglycidyl ether (EGDE) as a crosslinker. Firstly, MCC was dissolved in NaOH/urea, and MCC-CMC and MCC-xylan precursor solutions were prepared by adding carboxymethyl cellulose and xylan to the reaction mixtures. Then EGDE crosslinker was added dropwise into the homogeneous mixture to afford the final MCC-based hydrogels (Figure 15).

The in vitro delivery of Cephalexin was carried out in various simulated body fluids, such as artificial intestinal fluid (AIF), phosphate buffer saline (PBS), and artificial gastric fluid (AGF). MCC-CMC can deliver Cephalexin individually at 86% in AIF, 98% in PBS, 15% in AGF, and 98% in consecutive buffers (AGF followed by AIF and PBS) [100].

Nazir et al. reported the cytotoxicity studies of aminated cellulose derivatives against normal fibroblasts, melanoma, and breast cancer applications. Firstly, MCC was mixed with p-toluenesulfonyl chloride to obtain tosyl cellulose (C_TOS_), in DMAc/LiCl and Et_3_N. A further step was then the reaction with hydrazinium hydroxide, diethylamine, and diethylenetriamine to afford 6-deoxy-6-hydrazide cellulose (Cell Hyd), 6-deoxy-6-diethylamine cellulose (Cell DEA), and 6-deoxy-6-diethyltriamine cellulose (Cell DETA), respectively (Figure 16).

The results obtained show that Cell Hyd, Cell DEA, and Cell DETA exhibited noncytotoxic activity up to 200 μg/mL compared to normal fibroblast cells, confirming their safety in medical fields. These results suggested that aminated cellulose derivatives could be potential candidates for cancer-inhibiting studies and tissue engineering applications [101].

Finally, Pavliňák et al. studied the MCC oxidation with plasma-chemical treatment in order to obtain 6-carboxycellulose, one of the most used cellulose derivatives in the surgery field. Generally, health products based on oxidized MCC are employed as local hemostatic with unique bactericidal and fully bioabsorbable effects. This treatment is based on atmospheric plasma discharge in a liquid medium leading to the oxidation of MCC [102].

## 5. Conclusions

Several reviews on the isolation, preparation, characterization and applications of pristine and modified MCC in different fields were recently published [7,83,103]. Most of them focus on MCC and modified MCC applications in many different fields. 

Nevertheless, all studies have reported the significance of biodegradable resources and their impact on eco-friendly composites. Consequently, cellulose’s chemical, physical, and biological pre-treatment from different sources, such as plants, algae, and bacteria, has gained significant attention in research.

Microcrystalline cellulose combines high thermal stability, excellent rigidity, high crystallinity index, and high density with odorlessness, renewability, excellent mechanical properties, and easy preparation. It is used in many fields, such as food, cosmetics, packaging, and pharmaceuticals. Furthermore, MCC is suitable for different chemical modifications thanks to a large amount of accessible active hydroxyl groups on its surface, allowing it to enhance its chemical and physical properties further.

Pre-treatment of MCC is the main step for the preparation of cellulose-based materials and subsequent functionalization. The most important recent processes are reported here, also considering that researchers are called to face drawbacks concerning the high temperature often required, the high energy demand and liquid loading, the cost of the equipment, and the ultimate need for solvent recovery and recycling.

The chemical derivatization of MCC with functional components leads to the formation of advanced materials with new or improved properties. Therefore, their functionalities and potential applications in specific fields can be expanded. This is possible by exploiting innovative and greener chemical processes in order to derivatize hydroxyl groups onto MCC surface, broadening the range of possible applications of this chemically modified biopolymer.

We hope the present review highlighted recent advances in MCC derivatization, and relevant applications of the synthesized products.

## Figures and Tables

**Figure 1 molecules-28-02009-f001:**
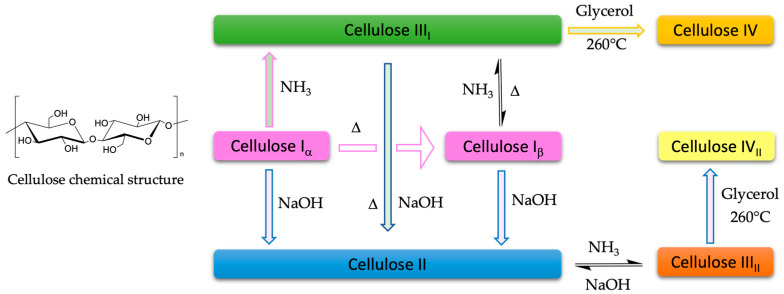
Diagram of the polymorph syhthesize of cellulose [12].

**Figure 2 molecules-28-02009-f002:**
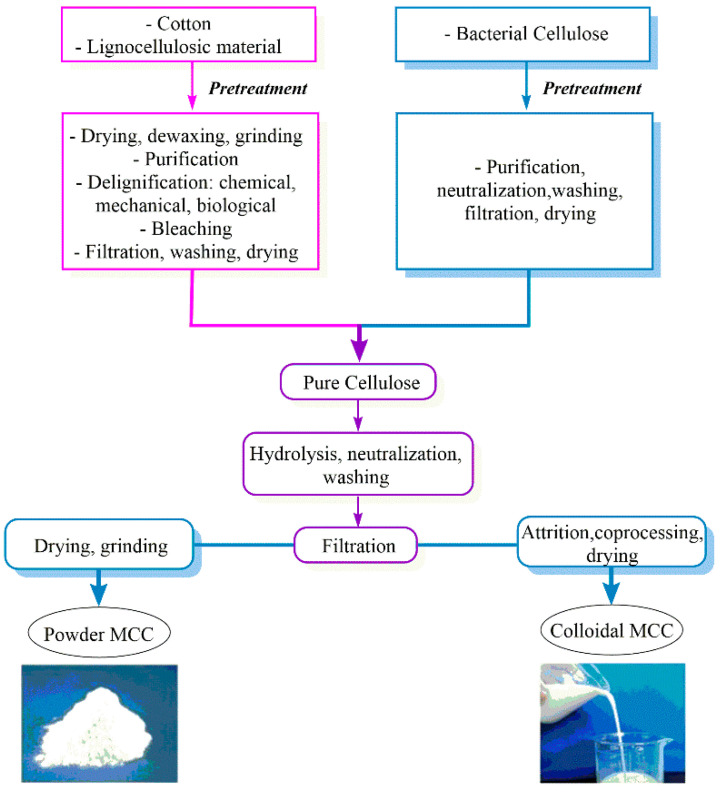
Production of MCC from cellulose-based materials [7].

**Figure 3 molecules-28-02009-f003:**
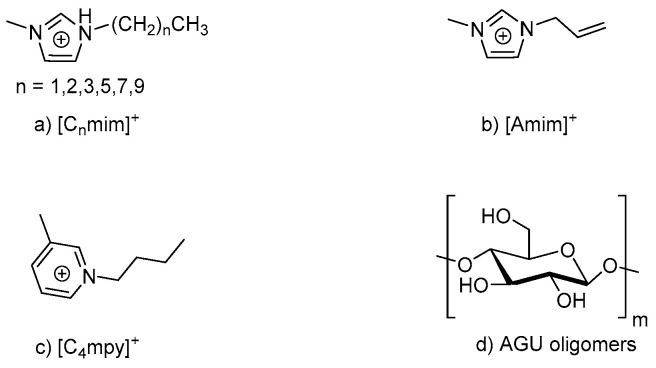
Structures for the ILs: (**a**) 1-alkyl-3-methyl imidazolium, (**b**) 1-allyl-3-methyl imidazolium chloride, (**c**) 1-butyl-3-methyl pyridinium chloride, and (**d**) β-anhydroglucopyranose (AGU) oligomers with a degree of polymerization of 10.

**Figure 4 molecules-28-02009-f004:**
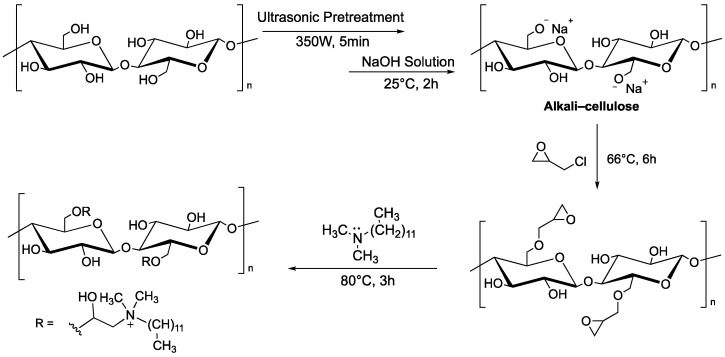
Two-step functionalization of MCC by reaction with glycidyl chloride, followed by ring opening with *N*,*N*-dimethyldodecylamine.

**Figure 5 molecules-28-02009-f005:**
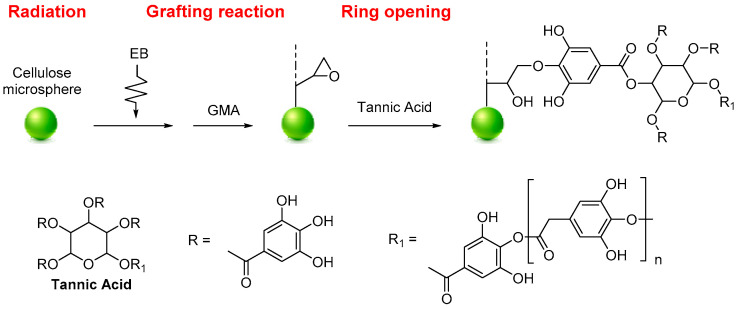
Synthesis of MCC-g-GMA-TA.

**Figure 6 molecules-28-02009-f006:**
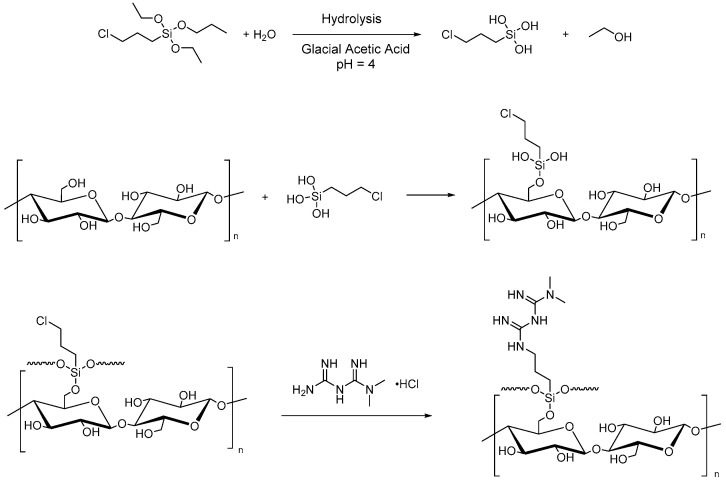
Silylation of MCC with CPTES and amine-functionalization (1,1-dimethylbiguanide hydrochloride).

**Figure 7 molecules-28-02009-f007:**
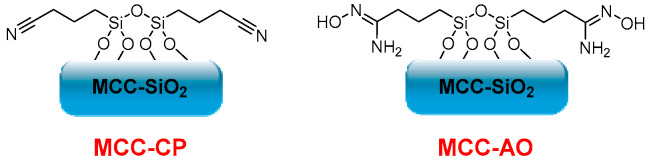
MCC-CP and MCC-AO structures.

**Figure 8 molecules-28-02009-f008:**
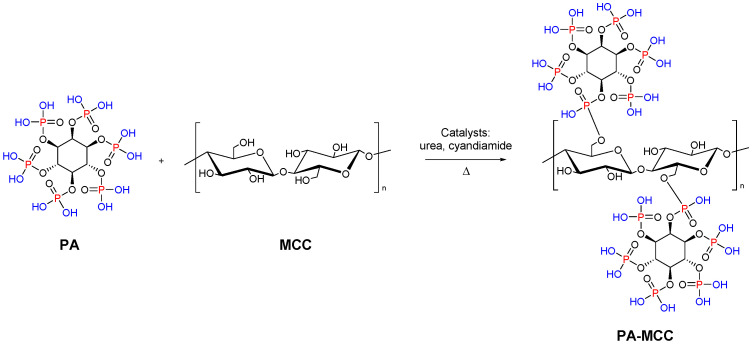
Synthesis of MCC modified with PA.

**Figure 9 molecules-28-02009-f009:**
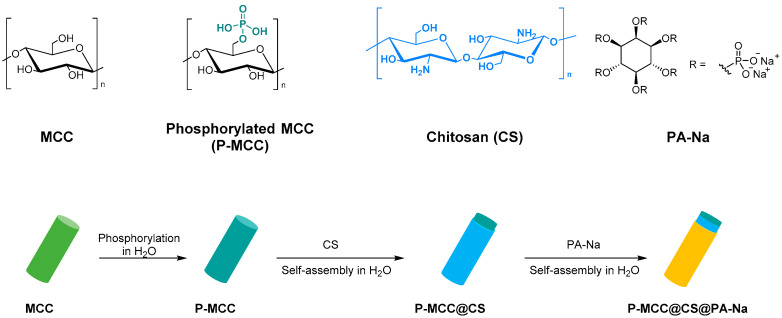
Synthesis P-MCC@CS@PA-Na.

**Figure 10 molecules-28-02009-f010:**
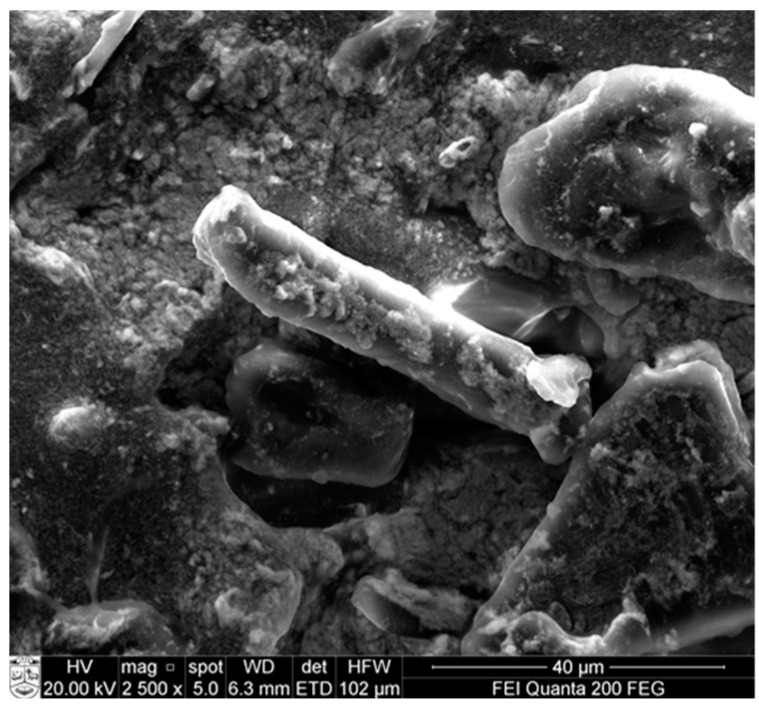
SEM image of PDMS/MCC composites with 20 wt.% of filler tensile fractured surface at magnification of 2500× and 20 KeV.

**Figure 11 molecules-28-02009-f011:**
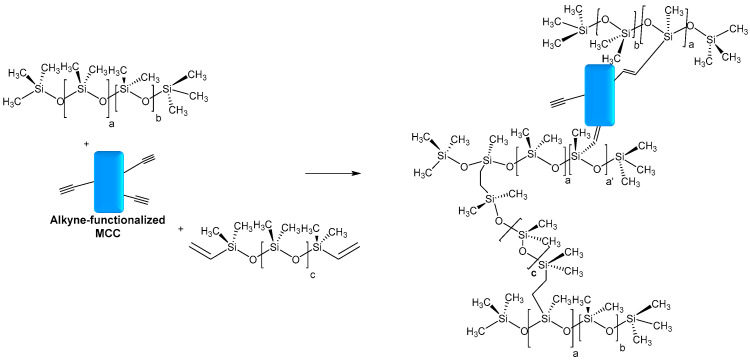
Possible hydrosilylation reaction occurring between alkyne-functionalized MCC and hydride-functional PDMS.

**Figure 12 molecules-28-02009-f012:**
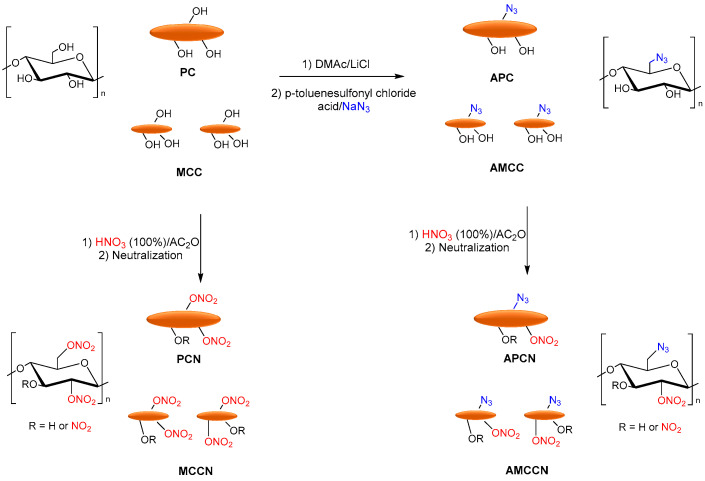
Synthesis of PCN, MCCN APCN, and AMCCN.

**Figure 13 molecules-28-02009-f013:**
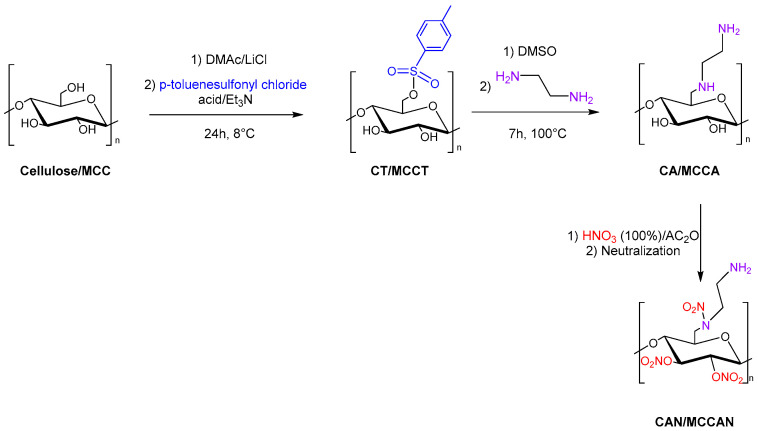
Synthesis of CAN and MCCAN.

**Figure 14 molecules-28-02009-f014:**
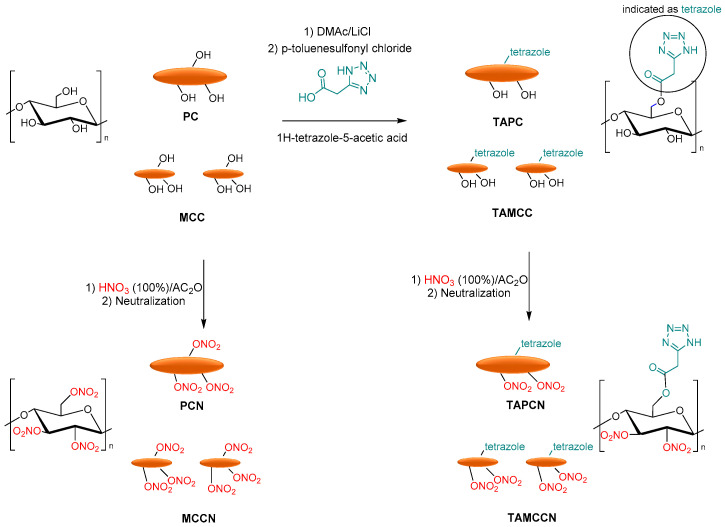
Synthesis of PCN, MCCN, TAPCN, and TAMCCN.

**Figure 15 molecules-28-02009-f015:**
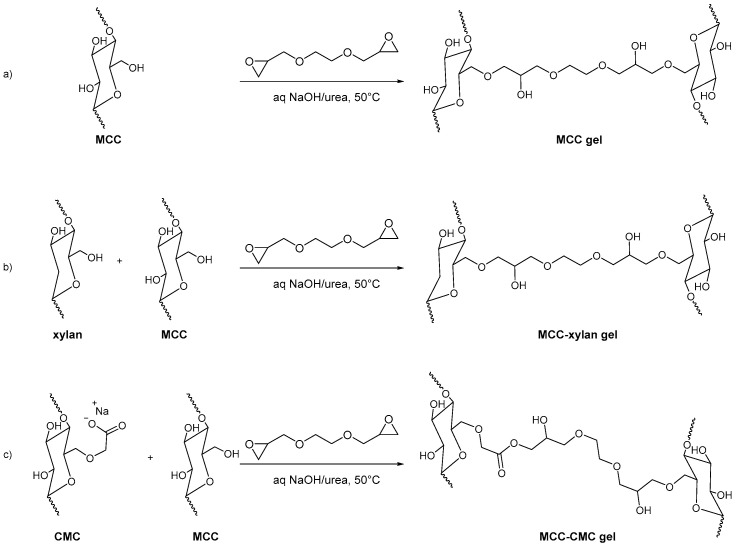
Hydrogel formation: (**a**) MCC gel; (**b**) MCC-xylan gel; (**c**) MCC-CMC gel.

**Figure 16 molecules-28-02009-f016:**
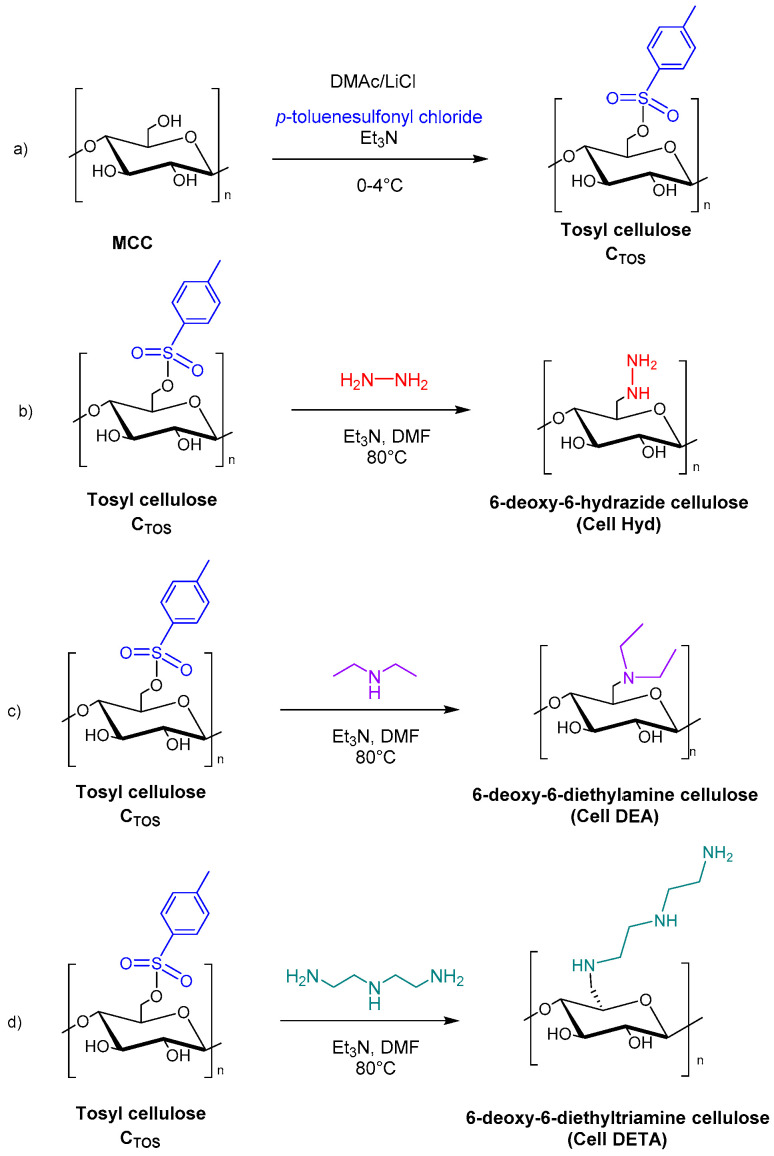
Synthesis of microcrystalline cellulose (MCC) derivatives: (**a**) tosyl cellulose (C_TOS_); (**b**) 6-deoxy-6-hydrazide cellulose (Cell Hyd); (**c**) 6-deoxy-6-diethylamine cellulose (Cell DEA); (**d**) 6-deoxy-6-diethyltriamine cellulose (Cell DETA).

## Data Availability

Not applicable.

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
