# Peer review of "Recent Developments in Chemical Derivatization of Microcrystalline Cellulose (MCC): Pre-Treatments, Functionalization, and Applications"

_molecules, 2023, doi:10.3390/molecules28052009_

Round 1
Reviewer 1 Report
The manuscript is devoted to a review of achievements in the production and modification of microcrystalline cellulose, as well as possible prospects for application.
This topic is relevant in view of the global trend towards the rejection of fossil carbon raw materials and the search for new polymeric materials of natural origin. The present review covers the achievements in MCC chemistry over the past twenty years. The review is well structured and written in an accessible language.
However, there are a number of disadvantages:
fig 2 improve quality
line 327-328 PDA or PDS?
line 350 GMA?
Authors should also try to correct the abstract, as in its present form, it does not describe the content of the review
You should also work with literature, because. there is a significant amount of literature published earlier than 2010, which is difficult to consider relevant today.
Also in the section devoted to obtaining MCC, I recommend the authors to consider the methods of oxidative delignification of wood, for example, the authors of Sudakova, Kuznetsov et al.
Reviewer 2 Report
molecules-2144250-peer-review-v1
Microcrystalline cellulose (MCC) recent developments: pre-2 treatments, functionalization, and applications
by Gabriele Lupidi
The authors review literature on preparation and application of microcrystalline cellulose from the past 20 years. What is the difference of this review to the already existing ones. A Google Scholar search with the search string “"microcrystalline cellulose" review” in the years 2021-2022 gave 2600 hits.
The “unique selling points” of this review must be elaborated more clearly before it can be accepted for publication.
There are some issues that must be reconsidered by the authors:
L8: The 1st sentence of the abstract is not true. Cellulose is the most abundant biopolymer but not MCC.
L22: Keywords are missing
L10: 1010 tons with superscript 10
L31: Acetobacter xylinum; species names should be typed in italics
L39: The term “anhydroglucose” is very common in literature but wrong: Levoglucosan is 1,6-anhydro-beta-D-glucopyranose. The repeating unit of cellulose is not anhydroglucose.
L26-L71: This is mostly textbook knowledge. It must be shortened to a minimum.
L80: How “virtually no weight loss” is achieved if amorphous parts of the polymer are hydrolyzed and washed out?
L103: “shorter reaction times” Than what?
L114: “alkali hydrolysis” is wrong in this context from my point of view. It is a oxidative degradation from the reducing end of the polymer chain. (caused by the added H2O2).
L178: 1 g to 1 kg (missing space between number and unit; g is the symbol for gram).
L188: High selectivity of atmospheric plasma must be explained in more details.
L239-L276: Every cellulose must be activated prior to dissolution in DMA/LiCl. According to own experiences, the dissolution process of MCC in any kind of solvent is much faster compared with cellulose of higher molecular weight. Reference 73 (Kosan et al.) does not explicitly describe MCC!
L292: The line break should be removed here.
L304-L313: It this information important here and for the use of MCC? As mentioned before, MCC dissolves faster, gives solutions of comparably low viscosity, which are necessary for the acquisition of nice NMR spectra.
Figure 3: The existence of “sodium cellulosate” (the sodium salt of the deprotonated cellulose) is questionable. It must be shown as sodium cellulose, where the hydroxide ion interacts with the proton of the OH group.
Figure 8: The structure of PA-NA seems to be wrong. If PA-Na is sodium phytate, then the ring oxygen atom must be replaced by COR.
L447: species names in italics
L451: …cellulose fibers from cotton linter were modified…
L468ff: Is there any advantage in case of MCC?
L491: The correct compound name must be used.
L517: “nitration of OH groups” is enough.
Figure 14: chemically correct compound names must be used.
Reviewer 3 Report
A review by Lupidi et al. focuses on microcrystalline cellulose (MCC), its obtaining, modification, and application. The authors review the possible crystalline states of cellulose, the obtaining of MCC both from cellulose-containing raw materials and through bacterial activities, and its production in the form of powder and aqueous dispersions. Then the authors discuss ways of physical, biological, and chemical treatments of cellulose. The authors mean that the chemical treatment of cellulose includes also its dissolution, and this is debatable because in most cases there is a direct dissolution of cellulose in solvents without the formation of new chemical compounds (cellulose derivatives) and therefore is rather a physical treatment. After that, the authors examine in detail the use of MCC and functionalized MCC as adsorbents, flame retardants, reinforcing agents, energetic materials, and biomedical agents. This list is not complete, as MCC has applications in other areas that would at least need to be listed. Nevertheless, the review is very well and clearly written, and therefore reads easily and with interest. In my opinion, it can be published after a revision.
Specific additional comments are as follows.
Line 8: “Microcrystalline cellulose (MCC) is one of the most abundant biopolymers existing in nature”. Microcrystalline cellulose is not found in nature. It is the product of processing naturally occurring native cellulose.
Line 9: “very limited fossil-based materials”. There are hundreds of fossil-based materials, in particular, many types of synthetic polymers with a wide variety of properties. What are their limitations? The authors should not indulge in wishful thinking.
Line 21: A replacement is needed: “and biomedical applications” -> “and biomedical additives”, “and biomedical agents” or “and in biomedical applications”.
Line 22: Keywords. The authors did not specify keywords.
Line 29: “to be over 7.5 x 1010 tons” -> “to be over 7.5 x 1010 tons”. The superscript style disappeared.
Figure 2 and many other figures. It is worth trying to increase the font size of the captions within the figures where possible.
Lines 148-215: As physical methods of breaking the crystal structure of microcrystalline cellulose, the authors mention ball milling, mechanocatalytic depolymerization, dielectric barrier discharge, and ultrasound technology. However, another way of transforming MCC into amorphous cellulose is its regeneration from solution by phase separation (see, e.g. 10.1021/bm3009022, 10.1007/s10570-021-04166-1). This method also deserves a mention, perhaps in the section on chemical methods on lines 270-292, although it is debatable because no chemical interactions during dissolution and subsequent regeneration of cellulose occur.
Lines 314-570: The authors review the use of microcrystalline cellulose as adsorbents, flame retardants, reinforcing agents, energetic materials, and biomedical agents. However, they do not consider such a typical application of cellulose as a thickening agent. Due to its biodegradability, cellulose is attracting attention to the production of biodegradable lubricants and greases on its basis. Microfibrillated cellulose, nanocrystalline cellulose, and cellulose derivatives are most commonly used in these applications, but microcrystalline cellulose is also applied (e.g., by its physical modification with organomodified montmorillonite, 10.1016/j.triboint.2020.106318). The area of cellulose-based lubricants is developing rapidly and deserves to be mentioned even briefly in the review. At least as a listing of other applications of microcrystalline cellulose before the conclusions.
Lines 517-587: The conclusions lack the authors' vision (their prognosis) of the future directions of MCC applications, a description of existing problems, and ways to overcome them in the future. If the authors could expand the conclusion in this direction, it would be an improvement.
Round 2
Reviewer 2 Report
molecules-2144250-peer-review-v2
Recent developments in chemical derivatization of microcrystalline cellulose (MCC): pretreatments, functionalization, and applications
by Gabriele Lupidi
The authors have addressed most of my comments, However, I still feel that a further revision round is needed.
L308-L311: Reference [81 Cellulose 2021, 28, 10203-10219.] reports on the preparation of cellulose nanoparticles by precipitation of the polymer from its solution N-methylmorpholine N-oxide monohydrate and dimethyl sulfoxide by adding nonsolvents. No further derivatization is carried out. Thus, this statement is wrong. Moreover, the statement “absence of bacterial cellulose treatment” is misleading because does not play a role in the cited reference.
Figure 3: I still have severe concerns with the “sodium cellulosate” shown here. Even it has been taken form another source, mistakes are spread without being corrected. The conversion is carried out in aqueous solution. Please have a look at the pKa values. The product should be named “alkali cellulose” (see CELLULOSE (1997) 4, 99–107).
However, I leave it to the editor’s decision whether or not to consider my comments.
